# Interpretable Analysis and Reasoning Enhancement for LLMs via Cross-Generation Reasoning Tree

## Abstract

Generating diverse reasoning paths by varying the context, such as demonstrations, prompts, instructions, *etc*, or sampling methods, such as top-k, top-p, beam-search, *etc*, and then selecting appropriate paths via majority voting or verifier-based strategies to enhance the reasoning capabilities of large language models (LLMs) is a commonly recognized approach. Although both different contexts and sampling techniques can generate diverse contents, using sampling methods alone does not significantly enhance the diversity of generation. Context variation, however, while fostering greater diversity in reasoning, can also introduce negative effects. It causes that switching contexts can not necessarily lead to proportional improvements in performance. Therefore, there is a need to investigate how context influences LLM generation and mitigate any adverse impacts. The primary challenge lies in the inability to conduct comparative studies once divergences occur in reasoning paths generated under different contexts. Specifically, once the predicted tokens at a given step differ, it becomes unclear whether subsequent tokens in the inference path are influenced by the context or the content already generated. In this paper, we propose a Cross-Generation Reasoning Tree (CGRT) algorithm for studying the impact of different contexts on LLM generation and enhancing LLMs' reasoning performance. Experimental findings reveal that, beyond enhancing interpretability, CGRT integrates the positive effects of both context and sampling strategies more effectively than previous approaches, leading to more rational inference paths. Experiments conducted on Llama2, Llama3, and Qwen demonstrate that, when generating an equivalent number of diverse inference paths, those produced via the "reasoning tree" method exhibit higher accuracy.

## 1 Introduction

Large Language Models (LLMs) have emerged with impressive performance in generative reasoning, often approaching that of human experts. OpenAI et al. (2023); Anil et al. (2023); Dubey et al. (2024); cla (2024) However, when confronted with more complex problems, LLMs may still make unexpected mistakes. A commonly recognized solution to this issue is generating diverse inference paths and then deciding the final answer via appropriate judgment strategies, such as "self-consistency" Wang et al.; Aggarwal et al. (2023) or "best-of-N" Liu et al. (2020), *etc*.

Typically, diversifying generated content can be achieved through appropriate sampling methods Fan et al. (2018); Holtzman et al.; Freitag & Al-Onaizan (2017); Chuang et al. (2023). Nevertheless, the diversity of sampling via decoding strategies remains constrained by the model's own distribution for answering the query question, which means that the diversity of sampled paths is limited when the model is highly confident. Besides, varying the context can also produce more diverse reasoning. However, inappropriate contexts can overly change the model's original distribution, sometimes resulting in significant negative effects. Therefore, it is necessary to investigate how context influences LLM generation and to correct any adverse impacts, which is a key topic that requires urgent attention in the field of LLM reasoning.

Often, conditioned with different contexts, once the LLMs' generation diverges at a certain token, the subsequent generated tokens cannot be compared. Therefore, previous interpretability analysis methods can only study the first inconsistent token or a few unexpected tokens but cannot track the entire reasoning process, *i.e.*, they cannot analyze the entire generation process under different contextual scenarios. Starting from the first inconsistent token, the subsequent predicted tokens are influenced not only by the context but also by the previously generated contents. To address this issue, we propose the "Cross-Generation Reasoning Tree" (CGRT). For each reasoning problem, CGRT constructs the LLMs generation into a tree structure. Each node in the tree represents a token generated by the LLM at a certain step. CGRT determines the next step of generation by placing different contexts before and after the problem, *i.e.* .few-shot chain-of-thought (CoT) Wei et al. (2022) demonstrations, prompts Kojima et al. (2022); Sahoo et al. (2024), instructions Efrat & Levy (2020); Mishra et al. (2022), *etc*, and then selects the child nodes of the current step node via certain sampling strategies, such as top-k, top-p, beam-search decoding, *etc*. When the generation at the current step diverges, the tree structure branches out. For each branch path of the tree, when determining the child nodes of a certain node, *i.e.*, the next step of generation by the LLM, the full combination of contexts and sampling strategies is applied. An example of the CGRT is illustrated in Figure 1. Thus, each branching point in the CGRT structure represents a divergence caused solely by the context, as the preceding generated reasoning path is controlled.

Through experiments using the Cross-Generation Reasoning Tree (CGRT), we further confirmed the widely acknowledged fact that context has both positive and negative effects on the LLMs' reasoning. Additionally, we conducted further investigations and found the following:

1. In CGRT constructed entirely from good contexts, a significant number of erroneous reasoning paths exist. Similarly, in CGRT constructed entirely from bad contexts, there are also correct reasoning paths.

2. The critical branching nodes in CGRT that determine the correctness of reasoning are predominantly composed of tokens with strong semantic information, such as numbers, operators, nouns *etc*. In contrast, words with weaker semantic significance, such as prepositions, conjunctions, punctuation, pronouns *etc*, are often not the critical branching nodes that determine the correctness of reasoning.

where good context refers to the context that, when placed before the question and decoded using greedy decoding, results in a correct reasoning answer; bad context, conversely, leads to an incorrect reasoning answer under the same conditions. Critical branching nodes represent nodes in a CGRT where all paths following the branching point lead to correct (or incorrect) reasoning answers, while their sibling nodes lead to the opposite, i.e., all paths following its sibling nodes lead to incorrect (or correct) reasoning answers.

However, for real-world reasoning problems, it is challenging to determine which path to take at a branching point. From the perspective of a branching point, although some critical branching nodes determine the correctness of subsequent reasoning paths, this determination is based on the CGRT constructed by the current context combinations. Exhaustively enumerating all possible contexts to generate an exceptionally large CGRT is obviously impractical. Furthermore, from a semantic perspective, most critical branching nodes cannot be fully understood as to why they determine the correctness of reasoning when viewed only from the current step. Human experts would find that, from the token of critical branching nodes that lead to wrong answers, it is possible to reason towards the correct answer. However, the performance of LLMs does not reflect this. The most likely explanation is that LLMs remain biased causal language models. LLMs do not possess genuine reasoning capabilities but rather model human f. Due to factors such as training data and parameter scales, they are unable to model the reasoning embedded within human language completely.

To address the challenge of selecting the appropriate path at branching nodes, we propose the inference version of the Cross-Generation Reasoning Tree (CGRT), referred to as iCGRT. iCGRT draws inspiration from the majority voting strategies but applies it at both the token-level and path-level. When studying the interpretability of LLMs, CGRT typically selects a limited number of context combinations (usually $2-4$), and a small number of sampling tokens for each context combination (typically one token decoded via greedy strategy or $1-2$ tokens sampled via top-k/p strategy). In contrast, iCGRT selects a larger number of context combinations ($\leq 8$) and usually generates $4-8$ samples for each context combination. At each node, if the generated tokens across all samples

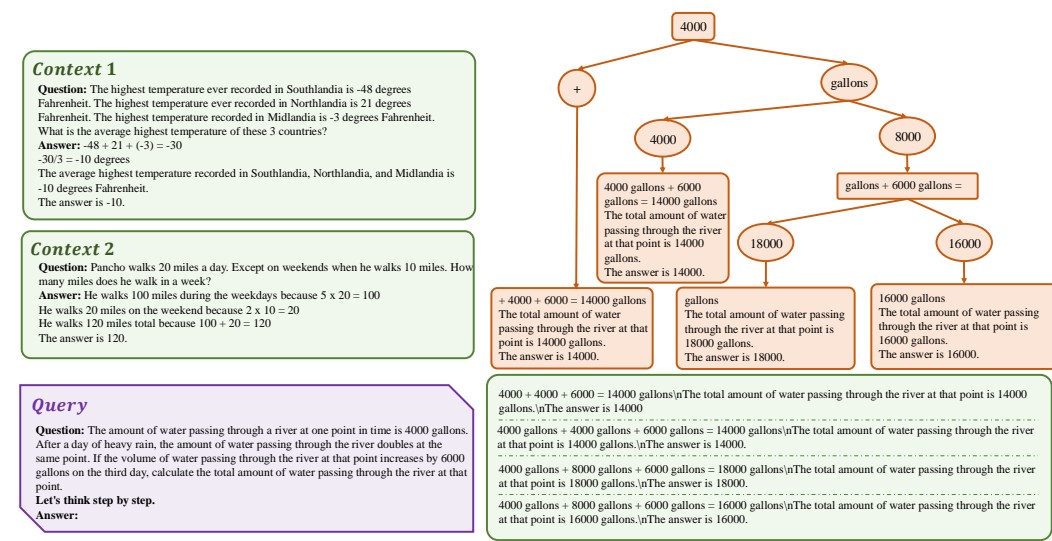

Figure 1: An Example of Cross-Generation Reasoning Tree (CGRT).

are inconsistent, a majority voting strategy is employed to select the top 2 predicted tokens as the nodes for that step. This approach has two benefits: firstly, the tokens selected through majority voting reduce the probability of leading to incorrect answers at that node; secondly, by restricting iCGRT to a binary tree, the computational burden of reasoning is reduced. Finally, the reasoning answer is chosen from all the reasoning paths in the iCGRT through a majority voting strategy. Experiments conducted on mainstream open-source LLMs, such as Llama2 Touvron et al. (2023), Llama3 AI@Meta (2024), and Qwen Team (2024), demonstrate that under the same number of context combinations and sampling quantities, iCGRT can produce more accurate reasoning paths.

## 2 METHODOLOGY

In this section, we first elaborate on the construction algorithm of the Cross-Generation Reasoning Tree (CGRT) after defining the mathematical notation. Subsequently, we present the interpretability conclusions of LLMs derived from CGRT experiments. Finally, we show the algorithm details of the inference-version CGRT, iCGRT.

### 2.1 MATHEMATICAL NOTATION

A N-ary tree can be represented as the pair of node set and edge set $\mathscr{T} = \{\mathcal{V}, \mathcal{E}\}$, where $\mathcal{V}$ is the node set and $\mathcal{E}$ is the edge set. The element of edge set, $(\mathbf{u}, \mathbf{v}) \in \mathcal{E}$, $\mathbf{u} \in \mathcal{V}$, $\mathbf{v} \in \mathcal{E}$, represents that the node $\mathbf{v}$ is one of the children node of the node $\mathbf{u}$. We define the insert operation as:

$$\mathrm{Insert}(\mathscr{T}, \mathbf{w}, \mathbf{p}) = (\mathcal{V} = \mathcal{V} \cup \{\mathbf{w}\}, \ \mathcal{E} = \mathcal{E} \cup (\mathbf{p}, \mathbf{w}))$$

For a node $\mathbf{v}$, we use the corresponding non-bold italic $v$ to denote the token it represents. With the notation conventions for an N-ary tree established, we define the Cross-Generation Reasoning Tree (CGRT) for the language model. For a query question $q$ and a $\mathcal{C}$ is the set of contexts, define the CGRT as $\mathscr{T}_{q,\mathcal{C}} = \{\mathcal{V}, \mathcal{E}, \mathcal{C}\}$. Each node represents a token generated at a certain step, and each node can have up to $|\mathcal{C}|$ child nodes, where $|\mathcal{C}|$ is the number of elements in the context set.

### 2.2 CROSS-GENERATION REASONING TREE

When generating the $i^{\text{th}}$ token, the LLMs' prediction distribution is influenced by three factors: context (such as demonstrations, prompts, instructions, *etc.*), the query question, and the content

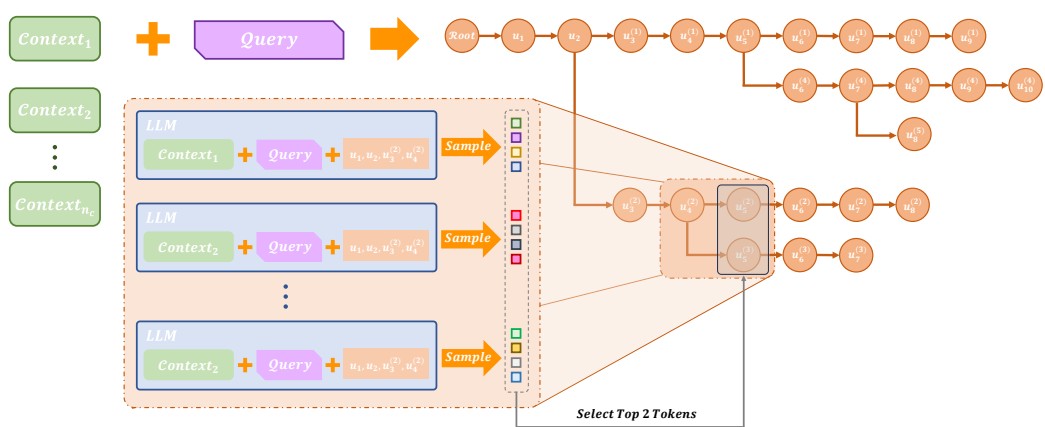

Figure 2: Figure 1: Illustration of the iCGRT construction algorithm. For each node in the iCGRT, when generating the next step's tokens, it considers $n_c$ contexts, and takes $n_s$ samplings for each context, generating $n_c \cdot n_s$ candidate tokens. If the candidate tokens are not entirely consistent, the top 2 tokens among them are selected as the child nodes of this node, thus limiting iCGRT to a binary tree. In practice, there are additional implementation details of iCGRT not illustrated in this figure; for a full explanation, please refer to section 2.3.

already generated. Under different contexts or samplings, or a combination of both, when divergence occurs in two inference paths from a certain step, the subsequent reasoning paths will appear different, although semantically they may convey the same meaning. Previous analyzing methods to seek LLMs' interpretability, such as probing Belinkov (2022), information flow Abnar & Zuidema (2020); Wang et al. (2023a), could only study the first branching position or manually identify locations in the reasoning path where unexpected errors occur. In order to delve deeper into the impact of different contexts on content generation, we propose the CGRT (Cross-Generation Reasoning Tree) structure.

For each query, given two contexts, CGRT constructs the model's generation into a tree structure. Specifically, CGRT represents each step of the LLM's prediction as a node in the tree, with each branch corresponding to a reasoning path. If the predicted token is <|end_of_sentence|>, the node is marked as a leaf node. For each step, one of the non-leaf nodes serves as the parent node. If the model predicts the same token for the current step under both contexts, a single child node is inserted for the parent node. If the predictions are inconsistent, two child nodes are inserted, indicating a bifurcation in the tree at this step. This process is recursively iterated until all terminal nodes in the tree are leaf nodes. In the case of binary CGRTs constructed from two contexts, only the paths of the leftmost and rightmost branches are generated based on information from a single context. Most other paths in the CGRT combine information from both contexts, which is the most significant difference from traditional tree structures and is the reason why we name the algorithm as a "Cross-Generation" Tree.

The formalized algorithm for constructing CGRT (Cross-Generation Reasoning Tree) is presented in Algorithm 1. It is important to note that the flexibility of language permits the same concept to be articulated in numerous varied manners. Hence, if no limitations are applied to the CGRT, this variability can lead to an extraordinarily high number of branches in some instances, potentially extending the construction time of a CGRT to several days. In practice, therefore, we opt to disregard branch point tokens that possess lesser semantic significance. The precise definitions of these non-significant tokens are delineated in Appendix A.1.4. If, at a branching point, some branches are determined to be non-significant, these will be disregarded; if all branches are deemed non-significant, a single branch will be chosen at random. However, even disregarding these tokens, the number of branches in CGRT might still grow uncontrollably for certain queries. Therefore, in practical implementation, it may be necessary to impose a limit on the maximum width of the tree.

---

**Algorithm 1:** CGRT (Cross-Generation Reasoning Tree)

---

**Input:** a language model $M$, a query question $q$, the set of contexts $\mathcal{C}$
**Output:** CGRT $\mathcal{T}_{q,\mathcal{C}} = \{\mathcal{V}, \mathcal{E}\}$

1 **Initialize:** a non-leaf root node $\mathbf{r}$ with empty data, $\mathcal{T}_{q,\mathcal{C}} \leftarrow \{\mathcal{V} = \{\mathbf{r}\}, \mathcal{E} = \varnothing\}$
2 **while** *there are non-leaf nodes without child nodes in the tree $\mathcal{T}_{q,\mathcal{C}}$* **do**
3   Select a non-leaf node $\mathbf{v} \in \mathcal{V}$, $v \notin \mathcal{W}_{\text{eos}}$ without child nodes
4   Get the path from the root node $\mathbf{r}$ to node $\mathbf{v}$: $\text{Sequence}(\mathbf{v})$
5   For each context $c_i \in \mathcal{C}$, given the input $c_i + q + \text{Sequence}(\mathbf{v})$, obtain the model $M$'s
   predicted token of the next step: $u_i = M(c_i + q + \text{Sequence}(\mathbf{v}))$
6   Remove duplicates from the predicted tokens across different contexts, resulting in $j$
   (where $1 \le j \le |\mathcal{C}|$) unique tokens:

$$\{w_1, w_2, \ldots, w_j\} = \text{Set}(u_1, u_2, \ldots, u_{|\mathcal{C}|})$$

7   **for** $w \in \{w_1, w_2, \ldots, w_j\}$ **do**
8    **if** $w \in \mathcal{W}_{\text{eos}}$ **then**
9     The node $\mathbf{w}$ is marked as a leaf node.
10    **end**
11    **else**
12     The node $\mathbf{w}$ is marked as a non-leaf node.
13    **end**
14    Insert the node $\mathbf{w}$ as a child node of the node $\mathbf{v}$:

$$\mathcal{T}_{q,\mathcal{C}} \leftarrow \text{Insert}(\mathcal{T}_{q,\mathcal{C}}, \mathbf{v}, \mathbf{w}) = (\mathcal{V} \cup \{\mathbf{w}\}, \mathcal{E} \cup \{(\mathbf{v}, \mathbf{w})\})$$

15   **end**
16 **end**

---

### 2.3 INFERENCE-VERSION CGRT

Theoretically, after obtaining a complete CGRT, an advanced selector could be employed to make choices at the branching nodes of the tree, thereby navigating the correct reasoning path. However, in practical application, it poses an almost insurmountable challenge. Firstly, at the step of branching nodes, although some branching nodes within a given CGRT may lead to entirely correct or incorrect subsequent reasoning paths, it is a phenomenon limited to the current CGRT. Given that CGRTs can vary infinitely due to the countless combinations of contexts, exhaustively enumerating all possible contexts to construct an extraordinarily large CGRT is impractical. Secondly, from a semantic perspective, most critical branching nodes cannot be discerned by human experts to determine what factors decide the correctness or incorrectness of subsequent reasoning paths. Often, even when a branching node leads to all incorrect reasoning paths within a certain CGRT, human experts are still capable of reasoning correctly from the current step. We enumerate many such cases in Appendix A.3. Numerous studies Jin et al.; Valmeekam et al. (2024); Kambhampati (2024) indicate that LLMs lack genuine reasoning capabilities - they merely model human language, and this modeling is biased. Due to factors such as training data and parameter scale, LLMs are unable to fully model the reasoning embedded within human language. This explains why the performance of LLMs diverges from that of human experts, influenced as it is by contextual factors. Consequently, making the right decision at the branching nodes of a CGRT, viewed solely from the step at the branching node, remains a challenging task.

Therefore, during the practical inference process, a solution to this problem is required. To address this, we propose the inference version of CGRT, referred to as iCGRT (inference-version ). The algorithm of building iCGRT is elaborated on the Algorithm 2. Compared to the standard building CGRT algorithm, iCGRT introduces the following modifications to adapt to the inference demands:

1. Employing a larger number of context combinations and more diverse sampling than used in interpretability studies to avoid biases caused by insufficient sampling. This adjustment is reflected in Line 5 of Algorithm 2.

---

**Algorithm 2:** iCGRT (inference-version Cross-Generation Reasoning Tree)

---

**Input:** a language model $M$, a query question $q$, the set of contexts $\mathcal{C}$, the decoding function $S$, the number of sampling steps per context $n_s$, the maximum number of iCGRT branches $n_p$

**Output:** iCGRT $\mathscr{T}_{q,\mathcal{C},S}^{(i)} = \{\mathcal{V}, \mathcal{E}\}$

1 **Initialize:** a non-leaf root node $\mathbf{r}$ with empty data, $\mathscr{T}_{q,\mathcal{C},S}^{(i)} \leftarrow \{\mathcal{V} = \{\mathbf{r}\}, \mathcal{E} = \varnothing\}$

2 **while** *there are non-leaf nodes without child nodes in the tree* $\mathscr{T}_{q,\mathcal{C},S}^{(i)}$ **do**

3      Select a non-leaf node $\mathbf{v} \in \mathcal{V}$, $v \notin \mathcal{W}_{\text{eos}}$ without child nodes

4      Get the path from the root node $\mathbf{r}$ to node $\mathbf{v}$: Sequence($\mathbf{v}$)

5      For each context $c_i \in \mathcal{C}$, given the input $c_i + q + \text{Sequence}(\mathbf{v})$, obtain the model $M$'s $n_s$ predicted tokens of the next step via the decoding method $S$:

$$\mathcal{U} \leftarrow \{u_{i,1}, \ldots, u_{i,n_s}\} = S(M(c_i + q + \text{Sequence}(\mathbf{v})), n_s)$$

6      Select the top 2 tokens from $|C| \cdot n_s$ predicted tokens:

$$\{w_1, w_2\} = \text{Top-2}(u_{1,1}, \ldots, u_{1,n_s}, u_{2,1}, \ldots, u_{2,n_s}, u_{|\mathcal{C}|,1}, \ldots, u_{|\mathcal{C}|,n_s})$$

7      **if** $|\{w \in \mathcal{U} \mid w = w_1\}| > \lambda |\{w \in \mathcal{U} \mid w = w_2\}|$ *or* $\text{PathNumber}(\mathscr{T}_{q,\mathcal{C},S}^{(i)}) \geq n_p$ **then**

8         $\mathcal{W}_{\text{kept}} \leftarrow \{w_1\}$

9      **end**

10      **else**

11         $\mathcal{W}_{\text{kept}} \leftarrow \{w_1, w_2\}$

12      **end**

13      **for** $w \in \mathcal{W}_{\text{kept}}$ **do**

14         **if** $w \in \mathcal{W}_{\text{eos}}$ **then**

15            The node $\mathbf{w}$ is marked as a leaf node.

16         **end**

17         **else**

18            The node $\mathbf{w}$ is marked as a non-leaf node.

19         **end**

20         Insert the node $\mathbf{w}$ as a child node of the node $\mathbf{v}$:

$$\mathscr{T}_{q,\mathcal{C},S}^{(i)} \leftarrow \text{Insert}(\mathscr{T}_{q,\mathcal{C},S}^{(i)}, \mathbf{v}, \mathbf{w}) = (\mathcal{V} \cup \{\mathbf{w}\}, \mathcal{E} \cup \{(\mathbf{v}, \mathbf{w})\})$$

21      **end**

22 **end**

---

2. Introducing a majority voting strategy during the inference at each node (each step generated by LLMs). This serves two purposes: firstly, it maximizes the avoidance of undesirable token generation; secondly, since a large number of tokens (often $\leq 32$) are generated per step, leading to an overly extensive tree structure, retaining only the top-2 tokens limits the tree structure to a binary tree, thus significantly reducing the number of reasoning paths in the iCGRT. This modification is reflected in Line 6 of Algorithm 2.

3. Building upon the previous improvement, if the quantity of the top-1 token far exceeds that of the top-2 token (for example, if the top-1 token count is more than 4 times that of the top-2 token), we consider it as a non-branching node. It is controlled by the input hyperparameter $\lambda$ and the first condition in Line 6 of Algorithm 2.

4. iCGRT imposes limitations on the size of the tree to prevent issues arising from excessive branching in rare cases. Specifically, when the total number of paths in the tree exceeds a predefined maximum, only the top-1 token is selected during node generation. This limitation is governed by the input hyperparameter $n_p$ and the second condition in Line 6 of Algorithm 2.

In addition to the methods mentioned above for accelerating and preventing excessive computational costs, iCGRT can also process only half the number of tokens at each prediction step to approxi-

Table 1: *maj-vote acc.* represents the accuracy of the predicted answer via majority voting strategy, and *prop. of correct paths* represents the proportion of correct reasoning paths among the total reasoning paths. The experiment uses the Llama3-8B model on the GSM8K test set via the 1-shot CoT setting. For each context, the CGRT generates predictions via greedy decoding.

| num of context | 2 | | 4 | | 8 | |
|---|---|---|---|---|---|---|
| metric | maj-vote acc. | prop. of correct paths | maj-vote acc. | prop. of correct paths | maj-vote acc. | prop. of correct paths |
| hard level 1 | 20.83 | 14.57 | 23.31 | 15.28 | 28.65 | 19.81 |
| hard level 2 | 0.00 | 1.34 | 7.54 | 3.63 | 8.01 | 3.34 |
| hard level 3 | 0.00 | 2.75 | 6.89 | 3.22 | 5.04 | 3.19 |
| hard level 4 | 0.00 | 2.65 | 3.01 | 4.13 | 2.89 | 1.34 |

mately to accelerate the inference. Specifically, in Step 6 of Algorithm 2, only half of the $n_s \cdot n_c$ tokens that originally should be generated are predicted. If the first half $n_s \cdot n_c/2$ tokens are consistent or the number of top-1 tokens far exceeds the number of top-2 tokens, the generation of the remaining half is skipped. In practice, there is a considerable number of prediction steps where all $n_s \cdot n_c$ tokens are consistent. This approach can reduce the computational load by approximately $1/3$. Throughout this paper, unless otherwise specified, all iCGRT experiments employ this method to accelerate inference.

## 3 EXPERIMENTS

### 3.1 MINING CORRECT REASONING PATHS FROM BAD CONTEXT

By constructing the CGRT, we found that even when using bad contexts, *i.e.*, contexts that cause LLMs to answer query questions incorrectly, the CGRT still generates some correct reasoning paths. To quantify this phenomenon, we first define four criteria to measure the difficulty levels of samples:

A. Randomly select $N$ combinations of demonstrations, and generate one inference path per combination using greedy decoding. The answer derived from the majority voting of these $N$ inference paths is incorrect.

B. In addition to satisfying criterion A, all $N$ inference paths generated must have incorrect answers.

C. From the $N$ combinations of demonstrations mentioned in criteria A and B, randomly select $M$ (where $M \leq N$) combinations. Generate $S$ inference paths for each of these $M$ combinations using top-k or top-p sampling. The answer derived from the majority voting of the $M \cdot S$ generated inference paths is incorrect.

D. In addition to satisfying criterion C, all $M \cdot S$ generated inference paths must have incorrect answers.

With these four criteria, we categorize more challenging samples into four levels, ordered from less difficult to most difficult:

1. Hard Level 1 (Not Very Hard): Meets criterion A

2. Hard Level 2 (Normal Hard): Meets criterion B

3. Hard Level 3 (Very Hard): Meets both criteria B and C

4. Hard Level 4 (Extremely Hard): Meets both criteria B and D

Conventional self-consistency/majority voting strategies do not effectively address hard-level samples arbitrarily. Furthermore, criteria B and D involve even stricter constraints on the difficulty, requiring that none of the generated paths are correct. Therefore, for samples classified within Hard Levels 2-4, particularly those at Hard Level 4, it is challenging for models to produce correct inferences based on ordinary diverse sampling methods. We leverage CGRT to generate reasoning paths for these difficult samples, with the results provided in Table 1.

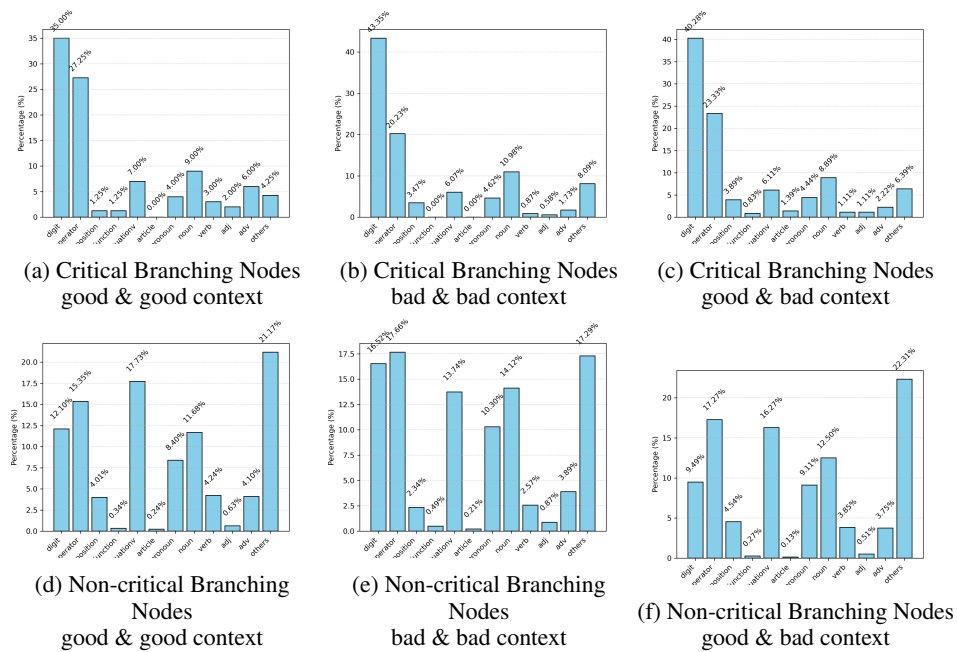

Figure 3: Part-of-Speech Analysis at Branching Nodes in CGRT, conducted on the GSM8K test set under 1-shot CoT setting via greedy decoding. For each query, we use two contexts to construct a binary CGRT. In the figure, good (or bad) context represents the model can reason correctly (or incorrectly) when we place it before the query.

From the experimental results, it can be observed that for challenging samples where majority voting strategy fails, CGRT is capable of predicting some of these samples correctly. Furthermore, for extremely difficult samples, *i.e.*, those for which correct inferences are not generated through normal sampling methods even after numerous attempts, although CGRT cannot predict a majority of correct inference paths, it does include some correct paths among its predictions. The reason can be roughly inferred to be that, CGRT employs a 'cross-generation' paradigm, allowing certain paths to benefit from multiple contexts. Due to the limitations of the model's capability, extremely difficult samples cannot be resolved with appropriate contexts or sampling methods, and in such cases, CGRT is similarly unable to generate correct answers.

In the experiments, to avoid excessive computational costs due to an unmanageable number of branches generated by CGRT for certain samples, we limited the maximum number of paths for CGRT to 16, 32, and 48 when using 2, 4, and 8 contexts, respectively. We observed that as the number of contexts increased, the accuracy of the paths generated by CGRT using majority voting also generally improved (although the cost of constructing CGRT also increased). However, it is noted that the metrics for 8 contexts is not always better than those for 4 contexts. This is because while an increase in the number of contexts indeed raises the probability of uncovering correct inference paths, it also leads to an increase in the number of incorrect inference paths generated.

## 3.2 PART-OF-SPEECH ANALYSIS AT BRANCHING NODES IN CGRT

We use two contexts for each query to construct a binary tree form of CGRT, using Llama3-8B model on the GSM8K test set with a 1-shot CoT (Chain of Thought) setting. Since the goal is to perform Part-of-Speech Analysis, we did not ignore the non-significant tokens defined in section 2.2 during the construction of the CGRT. The statistical results are shown in Figure 3. It can be seen that in mathematical reasoning tasks, the proportion of numbers and operators in critical branching nodes is higher, intuitively. However, because the figure only shows proportions and the number of non-critical branching nodes far exceeds that of critical ones, even if a branching node is a number or an operator, it is difficult to determine whether it is a critical one. In fact, if a branching node is a number or an operator, it is still more likely to be non-critical. In the appendix A.2.1, we illustrate

Table 2: Performance Comparison with SC Baseline

| Models\Dataset | GSM8K | | MAWPS | | SVAMP | | AQuA | |
|---|---|---|---|---|---|---|---|---|
| | SC | iCGRT | SC | iCGRT | SC | iCGRT | SC | iCGRT |
| Llama2-7B | 18.59 | **19.98** | **53.97** | 52.94 | **40.00** | 39.00 | **12.20** | 11.81 |
| Llama2-13B | 49.09 | **50.95** | 67.65 | **69.32** | 70.00 | 70.00 | 14.17 | **14.57** |
| Llama3-8B | 67.26 | **69.29** | 78.99 | **79.83** | 68.00 | **71.00** | 18.11 | **18.50** |
| Qwen2.5-7B-Instruct | 42.03 | **44.21** | 84.45 | **85.29** | 75.00 | **76.00** | **16.53** | 15.74 |

this point more clearly. In mathematical reasoning, parts of mathematical equations can adjust the order of computation, or the numbers or operators in the equation can be rearranged; for example, writing $16 - 3 - 4$ as $16 - 4 - 3$ is also acceptable. The part-of-speech analysis elucidates the complexity of branching nodes in the CGRT.

### 3.3 PERFORMANCE COMPARISON WITH TRADITIONAL SC BASELINE

We evaluate iCGRT against SC Wang et al. on four mathematical reasoning datasets: GSM8K Cobbe et al. (2021), SVAMP Patel et al. (2021), MAWPS Koncel-Kedziorski et al. (2016), and AQuA Ling et al. (2017), using four open-source LLMs: Llama2-7B Touvron et al. (2023), Llama2-13B, Llama3-8B AI@Meta (2024), and Qwen2.5-7B-Instruct Team (2024). The test results are shown in Table 2. All experiments are conducted using a 2-shot CoT setting. For each sample, two different examples are randomly selected as demonstrations. In the baseline experiments for SC, eight sets of 2-shot demonstrations are randomly selected for each query to serve as context. For each context, top-3 sampling is employed to generate four inference paths, resulting in a total of 32 inference paths. For iCGRT, the same eight sets of context are selected, and the same top-3 sampling method is applied to generate 32 tokens at each step (if the first 16 tokens is consistent or if the count of the top 1 token exceeded 12, the remaining 16 tokens will not be computed). The maximum number of paths for iCGRT is capped at $n_p = 32$.

## 4 RELATED WORKS

The paradigm of Chain-of-Thought (CoT) Wei et al. (2022) is commonly employed today to enhance the reasoning capabilities of models. Whether the manually or automatically constructed CoT demonstrations, or prompts Sahoo et al. (2024) or instructions Efrat & Levy (2020); Mishra et al. (2022) used to inform the model to think step-by-step, these can all be considered as context. The response given by LLMs to the same question can vary depending on the context. However, most current research on how context influences LLM generation remains at a relatively macro level, such as the length of the context Li et al. (2024), sequence order Chen et al.; Pezeshkpour & Hruschka (2024), domain conflicts Wang et al. (2023b), *etc*. A barrier that limits researchers from examining this issue from a more microscopic angle, such as at the token level, is that from the first inconsistent output token, subsequent generations will be jointly influenced by the context and the already generated content. Moreover, as language sequences progress, deviations become increasingly pronounced. Due to the polysemy of language, the same semantics can be expressed in many different ways. Therefore, finding a method to compare the reasoning content generated by LLMs under different contexts poses an exceptionally challenging problem.

This paper addresses the issue by constructing CGRT, a specialized tree structure with a cross-generative approach. Other methods that combine tree structures with Chain-of-Thought (CoT), such as ToT Yao et al. (2024), GoT Besta et al. (2024), XoT Ding et al. (2023), LLM+MCTS Zhang et al. (2024a;b), etc., differ from CGRT in two main aspects: Firstly, these methods use "thoughts" as units for tree nodes, whereas CGRT generates a tree structure at the token level. Secondly, the core objective of these methods is to enhance the model's reasoning performance, requiring the introduction of additional verifiers or evaluators to validate and assess the thoughts. In contrast, CGRT is a pure inference method that does not require an additional verifier. Instead, it leverages "cross-generation" to integrate positive influences from multiple contexts, thereby generating diversified reasoning paths with a higher probability of correctness.

Due to the frequent occurrence of hallucinations and logical errors in LLMs Huang et al. (2023); Xu et al. (2024), self-consistency Wang et al. is a widely adopted method to enhance the accuracy of LLMs' reasoning. The underlying insight is that LLMs may produce occasional errors in a single reasoning; thus, correct reasoning can be more possibly achieved through a majority voting on multiple diversely-sampled reasoning paths (by changing different contexts or utilizing varied decoding strategies Shi et al. (2024)).

## 5 DISCUSSIONS & FUTURE WORKS

**Inference Cost.** The inference speed of iCGRT is influenced by two factors: the number of tokens predicted per step ($n_c \cdot n_s$ in Algorithm 2), and the width of the tree (with the maximum width constrained by $n_p$ in Algorithm 2). Under all experiment settings presented in this paper, it is generally set that $n_c \cdot n_s = n_p$, and the SC baseline for comparison is also configured to generate an equivalent number of inference paths. If the number of tokens generated at each step matches that of the baseline inference path, and considering the cumulative length of the paths within the tree structure is typically longer than the average path length generated by the baseline, the computational cost would be higher than that of the SC baseline. However, iCGRT can avoid excessive computational burden by halving the number of tokens predicted at each step (as detailed in Section 2.3), which is satisfied during most generation steps. Additionally, the actual width of iCGRT often does not reach the predefined maximum width ($n_p$ in Algorithm 2) in many scenarios. Therefore, the computational overhead introduced by iCGRT is minimal.

**Future Works: More In-depth Research on the Impact of Context on LLMs' Generation.** As a tool of analyzing the impact of context on LLMs' generation at the token level, CGRT utilizes a "cross-generation" approach to eliminate the influence of previously generated content on the current token, attributing the branching nodes in reasoning paths solely to variations in context. However, the interpretability conclusions derived from this study remain preliminary, due to the following two challenges: Firstly, while CGRT can generate all possible reasoning paths for LLMs under different contexts for the same question, the polysemy of language means there are numerous ways to express correct or incorrect reasoning. This multiplicity leads to a large number of tokens non-decisive for the correctness of the reasoning. Future research should focus on identifying critical branching nodes within CGRT. Secondly, the topic of how context affects LLMs' generation is still acknowledged as a challenging subject, influenced by factors such as model size, pre-training, and post-training data, *etc*. Furthermore, whether LLMs merely represent powerful modeling of human language or possess genuine cognitive abilities remains a widely debated topic without definitive resolution Jin et al.; Valmeekam et al. (2024); Kambhampati (2024). Therefore, future works can focus on two aspects: On one hand, we can gain more profound insights into the interpretability of LLMs via CGRT. On the other hand, for the critical branching nodes of CGRT, efforts can be made to develop a robust token-level decision-maker to improve LLMs' reasoning performance.

## 6 CONCLUSIONS

In this paper, we introduced a novel algorithm, Cross-Generation Reasoning Tree (CGRT), aimed at enhancing the reasoning capabilities and interpretability of large language models (LLMs). The key challenge addressed is the impact of context variation on LLM generation, particularly the difficulties posed by diverging reasoning paths. CGRT tackles this by constructing a reasoning tree that tracks token generation across different contexts, enabling a thorough analysis of the role that context plays in model outputs. Our experimental results demonstrate that CGRT integrates the strengths of different context variations and samplings, generating more accurate and diverse reasoning paths. Moreover, CGRT reveals that even bad contexts can lead to correct reasoning paths and provides insights into the critical branching points that determine the success of reasoning. We also introduced an inference version, iCGRT, which uses majority voting at both token and path levels to improve decision-making at branching nodes, leading to more efficient and accurate reasoning in LLMs. In conclusion, CGRT offers a powerful tool for understanding and improving LLMs by combining multiple contexts and sampling techniques, enhancing both reasoning performance and interpretability. Future work could focus on deeper investigations into critical branching nodes and further optimizing the balance between reasoning accuracy and computational costs.

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

# A APPENDIX

## A.1 IMPLEMENT DETAILS

### A.1.1 DATASETS

GSM8K Cobbe et al. (2021) is a dataset of diverse grade school math word problems created by human problem writers. The test set contains $1,319$ problems. These problems need multi-step mathematical reasoning, usually taking between 2 and 8 steps to solve, and solutions primarily involve performing a sequence of elementary calculations using basic arithmetic operations ($+ - \times \div$) to reach the final answer.

MAWPS Koncel-Kedziorski et al. (2016) dataset is a collection of simple math word problems focused on arithmetic. Its test set contains 238 samples.

SVAMP Patel et al. (2021) is a challenge set for elementary-level Math Word Problems (MWP). An MWP consists of a short Natural Language narrative that describes a state of the world and poses a question about some unknown quantities. The test set contains 100 problems.

AQuA Ling et al. (2017) test set consists 254 algebraic word problems with natural language rationales. Each question provides 5 options and only 1 option is correct.

### A.1.2 MODELS

We perform experiments using the model Llama2-13B Touvron et al. (2023), Llama3-8B AI@Meta (2024) and Qwen2.5-7B Team (2024). We use the base models for Llama2 and Llama3, instead of the -chat or -instruct version. For Qwen2.5-7B, we use the -instruct version.

### A.1.3 FEW-SHOT COT SETTINGS

We use the following templates for the few-shot CoT inference:

```
Question: [QUESTION]
Answer: [RATIONALE]
The answer is [ANSWER].
Question: [QUESTION]
Answer: [RATIONALE]
The answer is [ANSWER].
...
Question: [QUERY]
Answer:
```

where `[QUESTION]`, `[RATIONALE]`, and `[ANSWER]` represent the question, rationale, and the final answer of the demonstrations respectively. `[QUERY]` represent the question to be reasoned.

### A.1.4 DEFINITION NON-SIGNIFICANT TOKENS

Due to that the flexibility of language permits the same concept to be articulated in numerous varied manners, we have to apply some limitations on the algorithm of building CGRT to avoid an extraordinarily high number of branches for some query question. we define the following tokens as non-significant ones. If, at a branching point, some branches are non-significant, these will be disregarded; if all branches are deemed non-significant, a single branch will be chosen at random.

```
non_significant_words = [
    'and', 'or', 'but', 'nor', 'for', 'so', 'yet', 'then', '
        therefore',
    'in', 'on', 'at', 'to', 'by', 'with', 'about',
    'a', 'an', 'the',
    'of', 'as', 'if', 'that',
    ',', '.', '!', '?', ';', ':', '_', '...', "'", '"',
    'what', 'which', 'who', 'whom', 'whose',
```

```
756      'this', 'these', 'that', 'those', 'he', 'she', 'it', 'there',
757          'here',
758  9   '\n', 'answer', 'question', '', ' ',
759      'am', 'is', 'are', 'was', 'were',
760  11  'per', 'every', 'each',
761      '$',
762  13  ]
```

## A.2 SUPPLEMENTARY EXPERIMENTS

### A.2.1 PART-OF-SPEECH ANALYSIS AT BRANCHING NODES

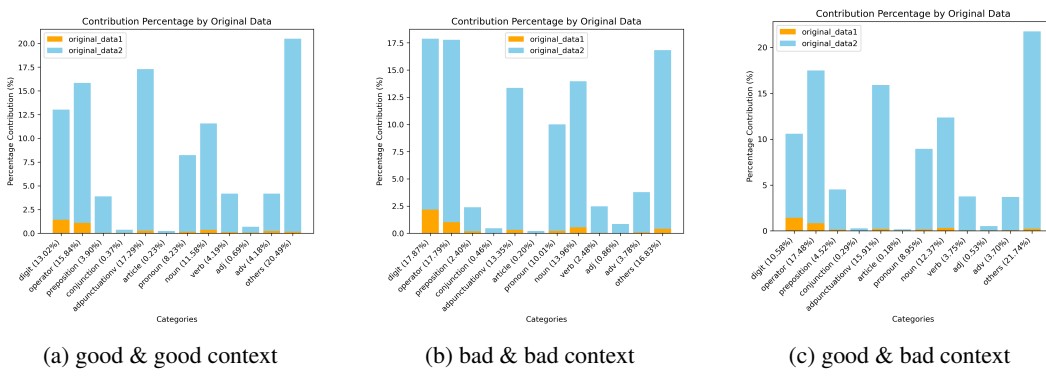

(a) good & good context   (b) bad & bad context   (c) good & bad context

Figure 4: Part-of-Speech Analysis at Branching Nodes

As Figure 4 shows, the quantity of non-critical branching nodes greatly exceeds that of their critical counterparts. Hence, despite the fact that within critical branching nodes, tokens that represent numbers or operators do comprise a significant portion, such an assertion cannot be reciprocally applied. Mathematical reasoning frequently encompasses non-serial logical computations that permit diverse permutations of operational hierarchies. Consequently, even in the CGRT, a branching node that represents a number or an operator cannot be easily identified as critical.

## A.3 EXAMPLES OF CGRT

Here we present some examples from the CGRT of the GSM8K test set. The symbols ·, −− , |, `
in the figures are used only for better visualization of the tree structure and have no actual meaning.
Because unrestricted CGRT often generates trees with a large number of branches, we have ignored
the branching nodes representing non-significant tokens.

And for better visualization, we have selected examples with an appropriate number and length of
reasoning steps. These examples are presented in the form of vector graphics (Figure 5) rather than
text.

Figure 5: Examples of CGRT.