# OpenReview forum: "Interpretable Analysis and Reasoning Enhancement for LLMs via Cross-Generation Reasoning Trees"
_ICLR.cc/2025/Conference — ICLR 2025 Conference Withdrawn Submission_

### Official Review · Reviewer_jksz · 2024-11-03

**Soundness:** 3
**Presentation:** 2
**Contribution:** 3
**Rating:** 6
**Confidence:** 3

**Summary:**

This paper introduces a Cross-Generation Reasoning Tree (CGRT) algorithm for improving the interpretability and reasoning capabilities of LLMs. The key innovation is representing LLM generation as a tree structure where each node represents a generated token, enabling analysis of how different contexts influence token generation and reasoning paths. The authors also propose iCGRT, an inference version that uses majority voting at both token and path levels to improve decision-making. The approach is thoroughly evaluated across multiple LLM architectures and mathematical reasoning datasets.

**Strengths:**

- A novel tree-based framework CGRT for analyzing how different contexts influence LLM generation at the token level
- Interesting findings include (1) incorrect contexts can still produce some correct reasoning paths and (2) non-semantic tokens rarely determine reasoning correctness

**Weaknesses:**

- While iCGRT includes optimizations, the tree structure can still lead to increased computational costs. Computational cost analysis and efficiency analysis will be beneficial to the experiments.
- Following the previous questions, for complex problems, controlling tree growth remains challenging despite the proposed optimizations.
- Experiments focus primarily on mathematical reasoning tasks - applicability to other planning benchmarks might be helpful to demonstrate the generalizability of this framework, especially in planning tasks with larger searching space?
- It seems to require careful tuning of hyperparameters like tree width limits and majority voting thresholds, any parameter studies for the robustness purpose?

**Questions:**

- Figure 5 is difficult to read.

---

### Official Review · Reviewer_tSFt · 2024-11-03

**Soundness:** 3
**Presentation:** 2
**Contribution:** 3
**Rating:** 5
**Confidence:** 4

**Summary:**

This paper proposes a novel method called iCGRT, inference-version of CRoss-Generation Reasoning Tree, to better prune the output paths generated from language models. Results on 4 reasoning datasets and 4 different language models show the effectiveness of the proposed method.

**Strengths:**

- The paper introduces an interesting phenomenon, when some tokens differ in the language model's output paths, the subsequent generations could be largely affected. Based on this observation, the authors propose to better prune the output paths for improved reasoning accuracy.

- The experiments are fairly comprehensive over four models, and four reasoning datasets.

**Weaknesses:**

- My major concern is that the proposed method, despite being much more sophisticated (involving many design choices as outlined in Algorithm 2), offers very marginal gains over the simple version of self-consistency. In some cases, e.g., SVMAP and AQuA, iCGRT is even worse compared to SC on some models. I'm not sure if the added complexity is worth the limited gains offered by iCGRT.

- In order to show the effectiveness/generalizability of iCGRT, can the authors provide more experiments on harder tasks? e.g., MATH, and other types of reasoning tasks, like commonsense reasoning and Big-bench?

- Can the authors provide ablation studies on the effect of n_s, n_p, S? How sensitive is iCGRT to those parameters / decoding methods?

**Questions:**

- Can the authors add more complex tasks (like MATH) and other types of reasoning tasks to the experiments?

- Can the authors provide ablation studies w.r.t. the hyper-parameters used in iCGRT?

---

### Official Review · Reviewer_D9b8 · 2024-11-04

**Soundness:** 3
**Presentation:** 2
**Contribution:** 2
**Rating:** 3
**Confidence:** 4

**Summary:**

This paper introduces Cross-Generation Reasoning Tree (CGRT), an algorithm that constructs a token-level tree structure to analyze and improve LLM reasoning across different contexts (e.g., in-context demonstration or instruction etc.) and sampling methods. CGRT can analyze the impact of context (i.e., independent of generations) to the reasoning correctness of LLM responses, by structure a tree structure for the inference path. The authors also propose an inference version (iCGRT), which restricts the width of the tree by taking majority votes at each node (reduce to binary tree). The authors perform experiments on mathematically reasoning datasets, and they show that good reasoning path can lead to wrong results and bad reasoning path can lead to correct results. They also compare iCGRT with self-consistent method and show improvement.

**Strengths:**

The paper proposes a new tree based decoding algorithm to analyze and improve reasoning capabilities of LLMs. This approach isolates the impact of the input prompt (or "context" as referred in the paper) from the already generated tokens to assess the correctness of the reasoning path. It provides insight into how context influence the LLM reasoning path through the analysis of critical branching nodes. Also, it adapts the algorithm to an inference version, which shows improvement on previously baseline method (self-consistency).

**Weaknesses:**

1. Limited insights from the analysis. I appreciate the attempt of the authors to propose a new algorithm to analyze the impact of the context to reasoning path of LLMs, however, beyond the algorithm itself I don't see much new insights from the analysis.  For example, one main finding from the paper is "good context can lead to incorrect answers and bad context can lead to correct answers,", this is not new and has been revealed from previous work (e.g., [1]). I would like to see the authors do more in-depth analysis with their method.
2. Lack of experiments. One of the main contribution claimed by the authors is the proposed methods leading to more accurate reasoning of LLMs, however, it is not well supported by the experiment:
    - The paper only compares with self-consistency method, but doesn't compare with other state-of-the-art baselines such as Tree of Thoughts or Graph of Thoughts.
    - The method improves over self-consistency (Table 2) but it is quite marginal (<=2%). Is that statistical significant? Even if so, how do we justify the significantly increased complexity introduced by the method (tree constructing and maintenance etc)? It is worth mentioning in the paper.
    - If the claim is about improvement of reasoning correctness on the reasoning path, there is no evaluation results to verify whether the reasoning path quality has improved.
3. I think the paper need improvement on the writing, here are a few examples:
    - Long sequences in the paper are not easy to follow. For example, line 13-17 in the abstract;
    - Fix the citation in line 62-64, and line 256.
    - Figure 3, it is not clear what is the difference between 3 plots on the same row. I think caption should be added to emphasize that.
    - As mentioned above, section 3.3 should be expanded to include more details, e.g., what metrics are used? How should we interpret the results?

reference:
[1] Language Models Don't Always Say What They Think: Unfaithful Explanations in Chain-of-Thought Prompting

**Questions:**

1. What will you do if the next token predicted by different context are evenly distributed (i.e., different)? This means low confidence from the model. How will you handle that?
2. Why the difficult level 1-4 are defined like that? Any intuition behind it? How do you determine "correct path"? Human? Also, this difficulty level is LLM dependent, are you doing the difficult classification per LLM?

---

### Note · Authors · 2024-11-18

I have read and agree with the venue's withdrawal policy on behalf of myself and my co-authors.